# Older Adults Affected by Abuse–What about Their Mental Health and Social Participation? A Mixed Methods Study

**DOI:** 10.3390/bs14030188

**Published:** 2024-02-27

**Authors:** Mari Salminen-Tuomaala, Juha Tiainen, Eija Paavilainen

**Affiliations:** 1School of Health Care and Social Work, Seinäjoki University of Applied Sciences, 60100 Seinäjoki, Finland; 2Finnish Student Health Service YTHS, 90570 Oulu, Finland; juha.tiainen@yths.fi; 3Health Sciences Unit, Faculty of Social Sciences, Tampere University, 33014 Tampere, Finland; eija.paavilainen@tuni.fi

**Keywords:** active and healthy living, informal social participation, mental health

## Abstract

The purpose of the study was to describe what types of abuse of older adults the healthcare providers in hospital emergency departments are currently able to identify. The study aimed at producing new information about the identification of abuse to enable the development of staff skills in the identification of abuse and in optimal interventions. The study is the first on the topic from the perspective of hospital emergency staff in Finland. The 76 participants represent 5 hospitals. The results are based on a statistical analysis of quantitative questions and on an inductive content analysis of participant experiences of suspected abuse. The inductive content analysis revealed that older adults subjected to abuse have narrowed social networks and many of them surrender to loneliness. Based on the relevant literature, the study discusses how the formal and informal social participation and mental health of this group of people could be promoted. Given the current limited resources, it is proposed that the idea of social prescribing might be applied informally, with help of a volunteer link person. Any interventions should be based on the older adults’ conceptions of what is meaningful to them.

## 1. Introduction

### 1.1. Definitions: Social Participation and Health

This mixed method study describes the situation of older adults affected by various forms of abuse in their own homes and asks how the objectives of social participation and mental health could be reconciled with their situation. The data concerning the situation of this group of older adults came from the emergency department staff of five hospitals in Finland.

It is ethically important to examine to what extent the situation of older adults affected by abuse is identified and how their situation is addressed in acute nursing. The services provided for this group of people can be regarded as a major indicator when evaluating the professional competence and educational needs of social and healthcare staff [1].

The staff in hospital emergency departments and in prehospital emergency services are in a unique position to identify abuse. Some forms of abuse may remain unidentified in hospital care, since part of the cues can only be detected in the person’s living environment [2,3]. It is important that social emergencies and abuse are recognized [4] and the social and healthcare staff provided with adequate up-to-date training and support [1].

Measures can be taken to prevent abuse at the level of society. Improving attitudes towards ageing and the older adults is one of the means. Multi-professional collaboration between health and social services should be increased, and public discussion on abuse should be encouraged [5]. The abuse of older adults should be addressed from a multidisciplinary and multiprofessional perspective, because any interventions call for coordinated collaboration between nursing and social care professionals [6]. In addition to addressing the acute situation, it is important that to support the mental health and social participation of older adults.

As for the definition of social participation, a consensual, interdisciplinary definition based on a scoping study was proposed in 2022: “a person’s involvement in activities providing interactions with others in community life and in important shared spaces, evolving according to available time and resources, and based on the societal context and what individuals want and is meaningful to them” [7] (p. 2).

According to Levasseur et al. [1], the definition involves a response to the questions of where, when and why. Referring to the work of Raymond et al. [8], Levasseur et al. note that the “why” entails such important dimensions of social participation as significant relationships, pleasant group activities, collective projects, and reciprocal support, assistance and sharing of knowledge. Social participation can also engender empowerment in decisions that concern older adults. It is voluntary and based on a conscious choice. It can be formal or informal and can, for example, take the form of volunteer work, leisure activities or informal support among families and neighbours. Many older adults today are active, healthy and conscious of their self-determination and right to make individualized choices. Still, ageing is often accompanied by illness, disability and restricted life space. For part of the older individuals, social participation may mainly mean interaction within the neighbourhood or through everyday activities such as shopping. In other words, social participation can mean very different things to different people. Not everybody can or wants to be involved in social activities according to normative expectations, and they should not be stigmatized [7].

What do the older adults themselves have to say about the topic? A recent study in Finland explored what constituted meaningful social participation from the perspective of older adults living independently in a non-institutional senior housing setting. For these individuals, meaningful social participation involved reciprocal practical and emotional support with fellow residents and family members; connecting with people they had something in common with; personal autonomy or freedom to decide for themselves; and being able to influence their living environment. These four aspects formed the overarching theme of feeling significant as a person; I matter [9].

In Finland, social participation of older adults has been formulated as an important objective in the law concerning the services for the older population [10]. The right to participate as autonomous citizens is also set out in two documents from the year 2020: the National Programme on Ageing 2030 [11] and the national-regional quality recommendation, whose aim is to guarantee a good quality of life and improved services for older persons [12]. The Finnish Institute for Health and Welfare, summarizing international research on social participation, suggests that the concept means being part of and contributing to communities and the common good, but also being able to choose and to determine how one lives [13].

For the concept of health, this study relies on Antonovsky’s well-known salutogenetic theory of the origins of health, which sees health and illness as a continuum regulated by a person’s sense of coherence. The sense of coherence is a person’s enduring confidence in the comprehensibility, manageability and, importantly, meaningfulness of life. Antonovsky’s theory involves the idea of generalized resistance resources, which facilitate the individual’s abilities to cope effectively with stressors [14].

Similarly, mental health is seen to exist on a continuum, as in a definition formulated by the World Health Organization [15]:

“Mental health is a state of mental well-being that enables people to cope with the stresses of life, realize their abilities, learn well and work well, and contribute to their community. It is an integral component of health and well-being that underpins our individual and collective abilities to make decisions, build relationships and shape the world we live in…Mental health is more than the absence of mental disorders. It exists on a complex continuum, which is experienced differently from one person to the next…”

### 1.2. Links between Social Participation, Health and Wellbeing

Research in social, health and behavioural sciences has confirmed that social participation, (mental) health and wellbeing are strongly associated [16,17,18,19,20]. The term mental wellbeing here refers to an individual’s personal experience of balance and wellbeing [21]. Social participation can help individuals remain active, while simultaneously promoting their health. It has been found to increase functional independence [22], to contribute to enhanced quality of life [23,24], to decrease mortality [25], psychological distress [26] and frailty [27] and to shorten hospital stays [28]. Supportive networks and neighbourhood cohesion and accessibility have been found to be factors facilitating social participation [29]. Group and community activities, a safe living environment and affordable public transport and physical activity opportunities have been stressed by the Finnish Institute for Health and Welfare as effective means of alleviating loneliness and social exclusion of older adults [17].

In contrast, lack of meaningful activities has been found to be a major barrier for social participation. What is meaningful to the persons themselves is likely to promote their health and wellbeing [30]. This means that those responsible for formulating health policies should remember to appreciate informal, non-standard forms of social participation as equal to the more formal or “mainstream” forms of participation [7,30]. Being able to cope with daily activities and attending to one’s needs are also thought to be determinants of social participation [31], and the person’s home is commonly seen as a place associated with independence and self-determination [7].

### 1.3. Examples of Interventions

A variety of interventions have been designed to promote the health, social participation and mental health of citizens, including older adults. In the United Kingdom, social prescribing is used in the National Health Service alongside medical treatment as part of personalised care to connect people to activities, groups and services in their community to meet their practical, social and emotional needs. Social prescribing means that GPs and other primary care professionals can refer people to a Link Worker, who assesses the situation and assists people in gaining access to their favoured local initiatives and projects provided by the voluntary, community, or social enterprise sector. Self-referral is also encouraged. This holistic approach can support older adults to deal with loneliness, improve their levels of physical activity and mental well-being, and help them take greater control of their health and wellbeing. Examples of the activities include volunteering, arts, group learning, befriending groups, cookery, lifestyle advice and physical activity [32]. At best, social prescribing can empower older adults to address various problems in their lives [33].

An example of the social prescription approach is the Creation POP project (2022–2024) with five European Union countries, including Finland. The project aims at promoting the wellbeing and social inclusion of older adults through creativity-based intergenerational activities (theatrical workshops). Another aim of the project is to expose stereotypes and prejudice on older adults and to increase understanding between generations [34].

In Finland, too, multiple interventions have been planned and implemented to promote the health, social participation and mental health of older adults, with emphasis on systematic collaboration, networking and reporting practices to detect older adults at risk of abuse [35]. A model piloted in Finland in recent years, funded by the Ministry of Social Affairs and Health and known by the acronym SOTE-TIKE, entails a coordinated effort to reach older adults with illness or social problems and to enable their independent coping at home longer. The interventions include coordinated social work, home care emergency services and crisis interventions brought to the older adult’s home [36].

A number of projects funded by the European Social Fund and local authorities have sought to promote the social participation and mental health of the citizens. Examples involve arts courses, gardening, cooking and shared meals. Part of the projects’ philosophy has been to provide appropriate facilities, for instance, village houses and common lounges in blocks of flats [37]. A current regional example involves psychosocial reminiscence groups organized by teachers and students of social work and health care and designed to improve older adults’ function, mental wellbeing and social participation [38].

### 1.4. Abuse as a Challenge to Social Participation and Health

Being subjected to some form of abuse in one’s home is a factor that can significantly reduce the social participation of older adults and lead to a number of serious health problems and other adverse effects. Abuse is here defined as by the World Health Organization as the violation of human rights, including physical, sexual, psychological and emotional abuse; financial and material abuse; abandonment; neglect; and serious loss of dignity and respect [39]. Based on a review of over 50 studies in 28 countries, it was estimated in 2017 that nearly 16% of adults aged 60 and older had been subjected to some form of abuse over the past year [40]. It should be kept in mind that much of the abuse remains unreported [41]. As an example, although 25% of older adults in a Swedish study reported having experienced some form of abuse in their lives, only 2% of had been asked questions about potential abuse within health care [42].

Research on the abuse of older adults involves reports on mental/emotional [43,44,45], social [44], physical [45,46], sexual [47] and financial [45] abuse. According to a recent study in Thailand, as the function of the older adults shrinks and they are forced to rely on others, the risk of abuse increases [48]. Older adults are also subjected to structural abuse in society. They may, for example, be discriminated against or denied the services they require or are entitled to [35].

Exposure to abuse in later life has been found to be associated with prolonged mental health problems, including depression, anxiety and suicidality, and with increased hospitalization and mortality risk [49,50,51]. For example, a study in Sweden found earlier experiences of abuse to be highly prevalent among hospitalized older patients [52]. As can be expected, prolonged abuse reduces life satisfaction [48,53]. Experienced abuse has also been found to increase the functional dependence of older adults [54].

### 1.5. Abuse of Older Adults in Finland

In Finland, an increasing number of older people continue to live in their own home despite illness and disability, supported by their families, volunteers and home care services [55,56]. It has been estimated that 16–30% of the over 65-year-old population have a mental health problem [57]. The estimates on the prevalence of abuse vary a great deal. The data available has been partly self-reported by home-living older adults [58,59], partly based on the observations of social and healthcare managers or professionals concerning older adults living in their own homes [6,60] or in institutions [61,62,63].

A prevalence study of self-reported abuse and violence among home-living older women from 2011 revealed that 25% of 60–97-year old women in Finland had been subjected to abuse or violence in the last 12 months [59]. In 2020, however, only approximately 7% of the 55–74-year-old women and 4% of men in the same age group in Finland reported that they had been subjected to some form of abuse [58]. A recent mixed methods study with prehospital emergency care providers in Finland provides a detailed glimpse into the lives and coping of home-living older adults, including an overview of the quality and safety of the informal or formal home care. The study revealed that many older adults had been subjected to psychosocial abuse, which involved belittling, verbal threats and other abusive communication from spouses, children or homecare professionals. The emergency care providers had witnessed signs of physical abuse—malnutrition, dehydration and lack of hygiene. Economic abuse, as well as self-neglect and self-abuse had also occurred. The term technological abuse was presented; the study showed that safety technology could sometimes be a source of anxiety to the older adults and only serve to provide an outward appearance of safety. The study introduced exhaustion, memory problems, jealously and alcohol problems as potential background factors to the abuse in the families [6]. Earlier research has indicated that carers’ stress, burn out and lack of support, knowledge and training are risk factors associated with the abuse of older adults [64].

National survey data collected from home care and nursing home managers in 2023 supports the notion that abuse is relatively common both in home care and in nursing homes. Nearly half of the managers (47%) in home care services and nursing homes had observed some form of abuse concerning their clients. In home care, 43% of the managers had reported family members or other close persons abusing older adults. In the nursing homes, it was most commonly the residents, who had engaged in abusive behaviour towards their co-residents. In addition, some of the homecare units (3%) and nursing homes (7%), reported staff’s abusive behaviour towards the clients [63]. An earlier study with data collected in 2011 from 50 round-the-clock care units in Finland revealed that psychological and social abuse—for example, neglect of wishes and requests—were the most prevalent forms of abuse, witnessed by a staggering 85% of the care staff. Physical abuse was less common, witnessed by 63% of the respondents [62].

Older adults are also affected by various forms of structural abuse. Currently too many older citizens have to wait for several months to receive various services or a place in a care home [65]. Mental health problems are often treated by medication only, without stopping to look at the individual’s overall situation [35].

### 1.6. Research on Social Participation of Older Adults Subjected to Abuse

A recent study in Thailand showed that mentally abused older adults who participated in social activities were happier than those who did not participate; the study revealed the stress-buffering effect of social participation on the older adults’ life satisfaction levels. In addition, the older adults living with their daughters were more satisfied with life than those who did not [48]. However, looking at the results of an extensive American study, one might ask if there are significant differences in patterns of formal and informal social participation and their association with the risk of abuse in older adults. The study revealed that individuals engaged in formal social activities (meetings, religious services and volunteer work) reported similar or higher levels of abuse than those with less frequent formal social participation, whereas women with regular informal contact with family members or friends were less likely to report abuse [66].

### 1.7. Study Purpose and Aim

In Finland, the abuse of older adults has been studied from the viewpoint of prehospital emergency care providers. This mixed methods study is the first on the topic from the perspective of healthcare providers of hospital emergency departments in Finland. The purpose of the study was to describe what types of abuse of older adults the healthcare providers working in hospital emergency departments are currently able to identify. The study aimed at producing new information about the identification of abuse at hospital emergency departments to enable the development of staff skills in the identification of abuse and in optimal interventions.

The study first describes how the healthcare providers see the situation of older adults affected by various forms of abuse in their own homes. The research questions were: What types of abuse of older adults do healthcare providers working in hospital emergency clinics identify? How easy is it for them to identify various types of abuse of older adults at their work? Secondly, based on the relevant literature, the discussion section asks how the social participation and mental health of this group of people could be promoted.

## 2. Materials and Methods

### 2.1. Study Design

This is an integrative mixed methods study [67]. The results are based on qualitative and quantitative data collected from hospital staff through an online survey. Mixed methods research is useful in studies on social phenomena and vulnerable populations [68,69].

### 2.2. Participants and Data Collection

The participants were 76 staff members (population 250, response rate 30.4%) from four 24 h central hospitals and one university hospital. In Finland (population 5.5 million), 12 of the 15 central hospitals and 5 university hospitals provide centralized and specialized 24 h services. The participants worked as registered nurses, practical (enrolled) nurses or paramedics in emergency departments. The paramedics had either a Bachelor’s degree in paramedics or a registered nurse qualification complemented by 30 ECTS credits of emergency care studies. The data were collected through an online survey tool called Webropol in autumn 2022 and spring–summer 2023. The participants did not take part in any workshop before the study was conducted.

In each of the hospitals, a coordinator of scientific research used the emergency department’s group e-mail address to send potential participants an e-mail with a link to a questionnaire. The cover letter contained an informed consent statement and detailed information about the purpose and use of the study. The participants were able to respond anonymously and they could not be identified in the research report. All staff members, irrespective of their age or sex/gender, were deemed eligible to participate.

The questionnaire contained eight background questions on the participants’ age, sex, basic and professional education, occupation, workplace and work experience in years. There were six quantitative Likert scale questions dealing with the frequency of suspected abuse in emergency departments, with the ease or difficulty to identify various forms of abuse and with the staff’s documentation and reporting skills. In addition, the questionnaire contained four open questions. The open questions dealt with the identification of abuse in emergency departments and with the emergency care providers’ needs for education. This article presents the results for the quantitative questions and for one of the open questions: “Describe the situations, in which you have identified signs of elder abuse in the emergency department”.

### 2.3. Analysis

The quantitative data was analysed using IBM SPSS Statistics for Windows 28. These results are displayed in frequencies, medians, means, standard deviations and percentages. The qualitative data were analysed using inductive content analysis [70]. First, the data were read through several times. Phrases, clauses and sentences, which represented an answer to the open question were picked up from transcribed data and reduced to shorter expressions while retaining the original idea. Reduced expressions with the same or similar contents were grouped to codes, which were combined into sub-themes and finally collapsed to the categories.

### 2.4. Ethics and Rigour

The study was conducted in accordance with the Declaration of Helsinki [71] and Guidelines of the Finnish National Board on Research Integrity [72]. Permission to conduct the research was granted by the management groups responsible for research in the five hospitals. In Finland, the research policy in hospitals is that an Ethics committee permission is only required when patients are involved as participants in a study.

The quantitative results on the identification of abuse can be considered generalizable, at least nationally. The sample can be considered representative, because it includes the staff of 5 hospitals out of the 12 hospitals that provide centralized and specialized 24 h services in Finland. Although located across Finland, the hospital represent regions with a relative uniform culture and population. The trustworthiness of the qualitative results was increased by the use of participant quotes and the investigator’s constant dialogue with the original data. In addition, saturation was reached as similar participant experience started to emerge [73].

The research team consisted of two female and one male investigator, two of whom had experience of emergency nursing. The investigators did not know the participants, who responded anonymously. A single investigator conducted the qualitative analysis, and the participants did not have an opportunity to comment on the findings, which can be seen to be limitations. However, two other researchers commented on the results of the analysis, which enhances the trustworthiness of the study. To increase the confirmability of the qualitative analysis, the analysis and results are described in detail, to ensure that the reader is aware of the strengths and weaknesses of the analytical process [73].

## 3. Results

### 3.1. Demographic Participant Data

Most, or 69 out of the 76 participants (response rate 30.4%), were registered nurses, but there were also 3 paramedics and 4 practical (enrolled) nurses. The majority of the participants were women (81.5%). The age range of the participants was 24–66 years (means 36.8 years; median 35 years). More than two-thirds of the participants (68.4%) had completed upper secondary education, and a great majority (82.9%) held a Bachelor’s degree from a university of applied sciences. Six participants had a Master’s degree from a university of applied sciences, four had a vocational qualification and three had a university degree. The participants’ work experience ranged from 2 to 40 years (mean 12.8 years; median 10.5 years).

### 3.2. Quantitative Data: Suspected Abuse among Older Adults as Reported by Emergency Department Staff

A great majority, or 70 out of the 76 participants (92.1%), had, in their work in the emergency department, encountered patients of various ages with suspected abuse. Over one-third of the participants (35.7%) reported that they had encountered such patients weekly. The other reports were monthly (32.1%); 1–3 times a year (23.2%); daily (5.4%) and over 3 times a year, but not monthly (3.6%).

The greatest group of participants (37.5%) reported that they had encountered “rather many” older adults with suspected abuse, whereas nearly as many (32.2%) had encountered “rather few”. More than one in five (23.2%) chose the more neutral option “neither many nor few”. In addition, 7.1% of the participants had encountered “very many” older adults with suspected abuse. Table 1 presents data on how easy or difficult it had been for the participants to identify various forms of abuse.

As shown in Table 1, physical abuse had been easier to identify than mental or social abuse. Nearly one out of three emergency care providers had found it “rather difficult” to recognize signs of mental or social abuse or neglect in older adults. However, when asked about their ability to document abuse, most participants assessed their skills as rather good (50%), very good (8.9%), or neither good nor poor (17.9%).

### 3.3. Qualitative Data: Descriptions of Suspected Abuse among Older Adults

Table 2 depicts the findings on the qualitative data. The inductive content analysis yielded three categories: the narrowed social participation of older adults; the neglected needs of older adults; and the violation of older adults’ rights; and human dignity. The table is followed by a more detailed description of the findings and complemented by direct participant quotes.

#### 3.3.1. Narrowed Social Participation of Older Adults

A narrowed social participation of older adults was one of the findings in this study, with 76 emergency care providers working in hospital emergency departments. According to the study participants, the narrowed social participation involved, first, surrendering to loneliness. It had seemed to the participants that the older adults had little influence over their loneliness; they had been left alone and lonely against their will. For example, they had shared their feeling that time passed slowly when there were no visitors and nobody to talk to. Their loneliness had also been visible in their appearance as melancholy, boredom and withdrawal. To quote some of the study participants,


*“The family members are conspicuously absent”; “There is nobody visiting the elders, nobody taking their hand and listening” and “They have this expressionless face, their surrender and neglect are visible in their eyes and posture” (p. 4).*


In the opinion of the study participants, their patients suffered for not being able to share meaningful things with other people. The older adults had told that “a lot of historical and tacit knowledge” was not being passed on, since nobody seemed to have the time or interest to listen. According to the study participants, this limited the sources of pleasure in the lives of older adults. In the words of one participant,


*The old people worry that nobody is interested in what they have done in their lives. Nobody needs the knowledge they could share (p. 13).*


Depressed older adults had often been left to cope on their own, according to the participants of this study. The participants assumed that their patients’ depression had been caused by their loneliness, deterioration of health, decreased mobility and loss of peers. Other indicators of depression involved malnutrition, loss of appetite and weight loss resulting in weakness, dehydration and electrolyte disorders. One of the study participants wrote,


*A cachetic old person with a rapid weight loss, no appetite and no family members taking interest. The EMS brought the patient haggard and hollow-eyed to the emergency department (p. 23).*


Secondly, the narrowed social participation of older adults referred to their lack of social contacts. According to the study participants, their patients had described how they had not been invited to family events so as not to cause embarrassment. To quote,


*An old person told me having been transferred from home for Christmas—having been replaced by the Christmas tree when the younger generation arrived to spend Christmas in the countryside (p. 54).*


On the other hand, the study participants pointed out that neglect could be unintentional. Sometimes families seemed to experience genuine distress and guilt for not having visited the old person more frequently. Signs of deterioration in the old person had often alerted family members to the need to acquire professional assistance and care.

Lack of caring family or friends had repeatedly brought older adults to the emergency departments. The study participants had witnessed several “revolving door” situations or readmissions to healthcare centres and hospital emergency departments, alternately. The call for an ambulance frequently came from home care professionals concerned about leaving the old person alone at home. Sometimes none of the healthcare organizations had seemed to assume responsibility for the overall situation. Below are some of the participant contributions to the topic.


*There are no family members living near to take care of these old people, they have been completely abandoned and they are brought to the ED in bad shape, neglected (p. 3).*



*Aged people are brought back and forth between home, the ED and the health care centre. They are often left in wet nappies without food and drink for a long time, as there is no time in the ED or anywhere else to care for them properly (p. 6).*



*Old people in too poor a condition in their homes, as the various actors close their eyes when there is no family (p. 17).*


Exclusion from friends following the deterioration of health was mentioned by the study participants as a significant factor in loneliness. The shrinkage of the social network could lead to the older adults neglecting their health and nutrition. In the words of one participant,


*When aged people leave their circle of friends, it starts to affect their mind and body, they do not care for their health anymore (p. 34).*


#### 3.3.2. The Neglected Needs of Older Adults

Families’ indifference towards factors that affect the coping of older adults was another result of the inductive content analysis in this study. The study participants had the impression that families were increasingly indifferent towards the physical and mental needs of older adults. The condition of older adults living alone in their homes was poorer than some years ago, and it was often the concerned neighbours, who called the emergency services. The study participants reported,


*The abuser is often a family member, and the abuse can be physical and social. They neglect the person’s basic needs, for example (p. 9).*



*There is physical violence from a family member and the old people are downright neglected (p. 65).*


According to the study participants, the older adults may also be cared for by exhausted family caregivers, who would also need help to be able to cope. For example,


*An aged person living at home with a family caregiver, and the care was so bad that the person’s skin was broken and the clothes were dirty (p. 2).*



*A mentally ill failed fledgling caring for his mother had given her a shove, so that she fell and broke her arm (p. 15).*


Another form of neglect involved families belittling health problems that affected the older adult’s coping at home. According to the study participants, some family members had tried to prevent the person telling about their situation in their own words, for example,


*A couple of times I’ve noticed families behaving strangely, interrupting the elder and belittling their concerns, in those situations I have intervened (p. 17).*



*There were often situations with patients, who had had several falls at home, but had not been taken to be examined—and no information about the causes of the falls. In these situations I sometimes reported my concern (p. 8).*


The participants reported that the neglect had sometimes extended to the safety of the older person’s environment. They had learnt about deficiencies in the physical, mental or social environment; thresholds that should have been removed to reduce the risk of falls, use of drugs or intoxicants among co-residents, prolonged loneliness and lack of social networks.

The older adults’ poor housing conditions were mentioned frequently in the study participants’ contributions. The causes were said to involve indifference, the family’s incapacity to deal with the problem, or lack of money; some families had been concerned about the hospital costs and had wanted to hide the old person’s modest home conditions. To illustrate these points,


*Mostly abandoned. The family’s indifference about how the elder can cope with daily activities. Basic home conditions that the family members have no interest in improving (p. 46).*



*For example not being able to cope at home, but the family members don’t want to see it. They got their blinkers on. Aged people may be admitted with bad personal hygiene and malnourished, but the family members still insist that they should be allowed to go home (p. 67).*



*Poor hygiene, poor heating and poor conditions in the home. No food in the home. Someone may have been appointed to deal with these things, but they don’t do anything (p. 3).*


Neglecting the older adult’s mental wellbeing could take various forms, in the experience of the study participants. The participants indicated that some families had belittled the fears, depression or anxiety of the older adults, or had not provided much emotional or social support. Instead of seriously considering the fears and their background as genuine experiences, the family members had “tried to play down” the fears or “disparaged the thoughts” of the older adults. To quote,


*The aged people say their families think that their fears are signs of hypochondria or senility (p. 19).*


According to the study participants, making light of depression or anxiety in older adults was a common reaction from family members, who seemed to consider mood disorders a normal part of ageing and a result of having to “give up many things”. The study participants pointed out that some families “did not realize that an aged person could have a meaningful life”. One of the participants wrote,


*I encountered an old person who had attempted suicide. The family members had not recognized the seriousness of the depression and were stunned about the suicide attempt (p. 76).*


It is possible, according to the study participants, that many family members, juggling work and life, simply fail to notice that the older adults are lonely and lack emotional and social support. The family may not be used to open communication or discussing topics like the need for support. To quote one of the study participants,


*They do not know how to consider the aged person’s emotions or need for emotional support, or the threshold to be crossed is too high for them to talk about it (p. 22).*


#### 3.3.3. The Violation of Older Adults’ Rights and Human Dignity

The final category, the violation of older adults’ rights and human dignity, involved the participants’ observations on families demeaning behaviour and lack of appreciation towards older adults and economic abuse of older adults. Giving orders or bossing around, speaking down on or threatening and pressuring the older adult were considered demeaning behaviour. During their patients’ short admissions to the emergency department, the study participants had witnessed demeaning verbal interaction, for example strong, impatient commands or patronizing language. Some older adults had been subjected to threats and pressure even in the presence of the study participants (nurses and paramedics). A quote from one of the participants:


*Adult children bossing around, commanding the aged person in a strong way (p. 42).*


Lack of appreciation towards older adults here means that according to the study participants, many family members ignored the will, wishes and opinions of older adults, or belittled their experiences. Families sometimes made quick everyday decisions from their own perspective only. It did not always seem to occur to them that the older adults might have different wishes. The study participants reported having heard:


*Mother doesn’t have personal wishes anymore, she’s so old (p. 33).*



*…can’t decide what is best anymore (p. 33).*


In addition, family members did not always take the person’s life experience or earlier experiences of illness into account; they made their decisions “based on the here and now”. Some study participants expressed concern over the violation of older adults’ rights to state their opinion. For example,


*The voice of the aged person is not heard, family members decide things for them (p. 4).*


Last, part of the older adults seemed to have been subjected to economic abuse. According to the study participants, their patients’ adult children may have used their patient’s money without their permission, or even denied them access to their own money. In an increasingly digital word, this can occur, when the older adult has no computer and has to depend on relatives to manage their bank affairs. As one of the participants wrote,


*Family members may be authorized users of bank accounts, but they don’t give the old person enough money, e.g., for food and medicines (p. 23).*



*…dirty, weather-inappropriate clothes. When asked, the aged person tells that the son takes care of the money affairs, and has not had time to give money for new clothes. Finally, the person admits having asked for money from their own account for several months (p. 1).*


Rather commonly, the study participants claimed that unemployed bachelor sons lived with their aged mothers, “getting liquor on their pension money”. Out of love or fear, parents do not report their adult children, even if it results in malnutrition for themselves. A participant described sad situations:


*Relatives and family members pressing for money, moving in with the aged person, abusing them, using intoxicants in their homes and threatening them (p. 45)*


## 4. Discussion

This mixed method study combined quantitative and qualitative data about the identification of abuse of older adults in hospital emergency departments. The quantitative data presents an overview of the situation of older adults indisputably or potentially subjected to physical, mental, social and economic abuse. Nearly 45% of the emergency department staff in this study had encountered older adults with suspected neglect or abuse. In this context, one must bear in mind that the identification of abuse other than physical is difficult. The alarming extent of abuse is corroborated by earlier studies, but the estimates vary significantly [6,58,59,60,61,62].

The qualitative data provides concrete examples, making the findings more transparent and enabling insight into the care providers’ experiences with older adults. On one hand, the study portrays a lonely person with no family or with an indifferent family; on the other hand, it depicts a person living with an abusive family member. The image in this study is of a rather stereotypical mother–adult-son situation, but other abusive relationships do exist, also outside the family context.

The findings contain a surprising number of detailed observations, which seems to indicate that the staff had adequate time resources. As the staff is obliged to report suspected abuse, they will probably have allocated extra time to talk to the older adults described in this study and to consult a psychiatric nurse or a crisis social worker. Still, the views presented represent subjective interpretations of the emergency department staff. The picture is not complete, since the older adults did not directly participate in the study. These points can be considered limitations to this study.

Abuse and neglect, combined with the stigma experienced by those affected by abuse can be considered major factors hindering the social participation of older adults and jeopardizing their mental health. Individuals subjected to abuse may shun contacts as a result of their guilt and shame, or an abusive family member may restrict their interaction with other people [6]. Suspected abuse should thus be addressed immediately, before considering other interventions, whether the abusers are family members, caregivers or other individuals associated with the older adults. It must also be borne in mind that the emergency admissions of older adults in poor health do not always indicate familial abuse; they may represent a form of structural abuse in society. This study revealed that at times, no institution seemed to assume responsibility for the older adults. It is officially recognized that many older citizens have to wait for several months to receive services or a place in a care home [65].

The situation calls for collaboration between health and social services. Systematic practices and uniform instructions on how to address suspected abuse could lower the threshold of intervening for health and social care professionals [41]. Steps have already been taken to integrate services and utilise the care capacity of a broader institutional and environmental set of actors [74]. Models based on coordinated services, such as the social prescription approach as part of personalized care in the National Health Service in the UK [32] or the SOTE-TIKE project in Finland [36] could be effective in the work with the older adults. The creation of a safety network, and sometimes the provision of emergency housing, is especially important for individuals affected by abuse.

Seemingly small steps could be decisive when seeking to promote the social participation and mental health of this group of older adults. With reference to Levasseur’s [7] definition of social participation: Who should ask what the older adults want and what is meaningful to them? Where and when could the social participation take place; is there a need to provide shared spaces? Are additional time and economic resources required, or is there already an adequate variety of social participation opportunities? Although the positive influence of social participation on mental health seems undisputed, individuals may be unwilling to leave their homes and participate in social activities. The problem is how to connect this group of older adults with the existing services. On the other hand, it must also be remembered that social participation can mean a lot of different things, sometimes outside normative expectations [7]. As this study showed, family members do not always realize that older adults could still have personal wishes and the right to experience their life as valuable and meaningful as in their youth—“I matter”, as formulated in a study with older adults [9]. As Viktor Frankl [75] noted, it is possible to maintain meaning in life in old age; health problems and other life changes do not automatically exclude the possibility of meaningful life.

Looking at the social prescribing model, one might ask if similar results could be achieved informally in the current times of limited resources. Could any person—a neighbour, relative, an acquaintance—with adequate interest, empathy and information seeking skills provide some of the services provided by link workers? An awareness/publicity campaign might be useful in introducing the idea to the public. An online and print campaign could possibly involve information and tips for the potential link person, combined with empowering, anonymous narratives of people coping with abuse. Both the older adult and the link person would need to be open to the idea that in some situations, people outside one’s immediate family can have an important role in effecting change. The attributes of an ideal link person would involve presence, listening and an encouraging attitude. In practice, the link person could ask a set of preset questions and seek information to connect the older adult with the favoured services, activities or people.

A potential result of such informal work could be re-establishing contact with a friend, neighbour, or relative, for example a grandchild. As noted by Levasseur et al. [7], the role of the immediate neighbourhood grows with the increasing age. A better relationship with a neighbour or relative could lead to reciprocal support, for example, when child care, transport or help with digital services is required. Another result for the older adults with a history of abuse could be to find peer support, which might be easier to accept than formal services, especially when feelings of guilt or shame are involved. Peer support could also be provided in the form of a book, film or leaflet, bringing the individual together with a “survival story” of a peer. Third, the link person might find out about the person’s previous interests, hobbies or dreams and explore if they could be revived. The link person should ideally be equipped with a number of tools. Reminiscence, for example with help of old photographs, could be used to help older adults reconnect with good memories, perhaps encouraging them to seek out old acquaintances or return to earlier favourite activities.

An important question is, of course, how to make this type of volunteer work worthwhile. For some individuals, it may be rewarding just to become aware of their potential role and be able to contribute to the wellbeing of older adults. Some volunteers might enjoy “becoming part of the narrative”, possibly wishing to share the story together with the older adult online, in print or in some other form. Some form of compensation or acknowledgement might help attract and maintain volunteers. Third, the viability of remote link work and remote peer contact might be considered. These questions reveal the need for further research and possibly pilot studies.

## 5. Conclusions

Older adults subjected to abuse have narrowed social networks and many of them surrender to loneliness. Suspected abuse should be addressed immediately. Following that, this group of older adults might benefit from both formal and informal forms of social participation. Given the current limited resources, it is proposed that the idea of social prescribing might be to some extent applied informally, with help of a volunteer link person. It is essential that the interventions are based on the older adults’ conceptions of what is meaningful to them and who they want to share it with.

## Figures and Tables

**Table 1 behavsci-14-00188-t001:** Ease or difficulty to identify various forms of abuse among older adults as reported by emergency department staff.

Form of Abuse	Very Difficult	Rather Difficult	Neither Easy or Difficult	Rather Easy	Very Easy	Mean	Median	Standard Deviation
For me identifying physical abuse among older adults is	1.8%	3.6%	30.3%	58.9%	5.4%	3.6	4.0	0.75

For me identifying mental abuse among older adults is	3.6%	30.3%	33.9%	26.8%	5.4%	3.0	3.0	0.93

For me identifying social abuse or neglect among older adults is	3.6%	28.6%	28.6%	33.9%	5.3%	3.1	3.0	0.93

**Table 2 behavsci-14-00188-t002:** Emergency department staff’s experiences of suspected abuse among older adults.

Codes	Sub-Themes	Categories
Experienced loneliness	Surrendering to loneliness	The narrowed social participation of older adults
Not being able to share meaningful things		
Depressed older adults being left to cope on their own		
No invitations to family events	Lack of social contacts	
Lack of caring family or friends		
Exclusion from friends following the deterioration of health		
Lack of interest in the older adults’ physical, mental and social needs	Families’ indifference towards factors that affect the coping of older adults	The neglected needs of older adults
Belittling health problems that affect the older adult’s coping at home		
Neglecting the safety of the environment		
Failure to deal with the poor housing conditions		
Belittling the fears of older adults	Families neglecting the older adult’s mental wellbeing	
Belittling the depression or anxiety of older adults		
Lack of emotional and social support		
Giving orders to the older adult	Families’ demeaning behaviour towards older adults	The violation of older adults’ rights and human dignity
Patronizing speech		
Threatening and pressuring the older adult		
Ignoring the older adult’s will and wishes	Families’ lack of appreciation towards older adults	
Belittling the older adult’s experiences		
Ignoring the older adult’s opinions		
Use of money without permission	Economic abuse of older adults	
Denying the older adult access to their own money		

## Data Availability

The data supporting the results reported are available in Finnish language from the researchers. There are no public links to the data.

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
