# Peer review of "Older Adults Affected by Abuse–What about Their Mental Health and Social Participation? A Mixed Methods Study"

_behavsci, 2024, doi:10.3390/bs14030188_

Round 1

Reviewer 1 Report

Comments and Suggestions for Authors

Thank you for the opportunity to review this interesting paper. I hope these suggestions are helpful.

Overall, this paper can be significantly strengthened by providing more specific content on the purpose/rationale of the current study and its explicit contributions to the literature/knowledge base and practice. It was surprising to me that knowledge about types of abuse elders in Finland experience in not known. Or if the issue is that knowledge about health care provider response is missing, then the additional information in the background and purpose would be of benefit.

Background: The background is broad and comprehensive in its discussion of elder abuse in Finland. It establishes that there is a documented problem. However, it is not clear if this is novel research. It is not made specific the problem/gap in knowledge that the current research hopes to impact/fill. I would suggest a finer point be made as to the rationale for the study, the gap in the evidence base that this research aims to fill along with its goals, aims, and specific research questions.  This be summarized at the end of the background section. This would enable assessment of the paper’s explicit scholarly contribution.

Methods:

Design – this section should provide additional information about the design approach and rational for the design and leave any purpose content to the background section.

Participants and data collection – At first, I was concerned that there was no information about how study participants were recruited/selected or ethical oversight (but see it later in ethics….would it be better placed in participants?  That is typically where this content is presented) Also, it would be important to include the potential population of nurses to understand the representativeness of the sample.

Further, it is difficult to assess the applicability and rigor of the survey questions as it is not made clear what the purpose is of the current research. Is it to understand what emergency room nurses see and need training in? And if so, has this never been explored before? Or is it something else like providers in Finland know it is happening but do not know how or what the impacts are that health care providers can inform?  

Ethics – the statement “The quantitative results on the identification of abuse can be considered reliable and generalizable at least nationally, because they represent five hospitals across Finland” is very problematic. There is nothing supporting this and 5 hospitals out of how many? And the unit of analysis seems to be individual nurses not hospitals. And reliability has to do with your measures (that are not documented as tested as reliable) and have nothing to do with representativeness or generalizability.

Results – The study results are clearly described.

Discussion – some over-stepping conclusions. This is a small survey and is not generalizable as asserted. More conservative language should be utilized.

Comments on the Quality of English Language

The English in this paper is awkward and could be improved through editing.

Author Response

Thank you very much for taking the time to review this manuscript. Please find our point-by-point responses below. The revisions have been marked in red in the re-submitted files. Please, see the attachment.

Reviewer 2 Report

Comments and Suggestions for Authors

 The paper is quite interesting and engaging for the readers. However, there are a few limitations that need to be considered. Firstly, no depression or anxiety scales were used in the study. Secondly, the research was conducted in five different hospitals, which may have led to confounding evidence due to cultural and psychosocial issues. Thirdly, the study did not discuss the significant role of socioeconomic factors. Finally, some of the study participants were younger and had less experience.

Can you please clarify if the participants underwent any workshop before the study was conducted?

           Please do explain the experienced abuse        

Comments on the Quality of English Language

In the introduction and conclusion, social prescribing is misspelled as social describing. Please correct it accordingly.

Author Response

Thank you very much for taking the time to review this manuscript. Please find our point-by-point responses below. The revisions have been marked in red in the re-submitted files. Please see the attachment.

Reviewer 3 Report

Comments and Suggestions for Authors

Dear, the topic of the paper is very interesting and I believe it is essential that research in this area is stimulated and valorised.

From a methodological point of view, it is necessary to integrate in chapter "2. Materials and Methods" some elements with respect to the indications provided in the COREQ Checklist or other specific checklist for mixed-method studies (e.g. Fetters, M. D., & Molina-Azorin, J. F. (2019). A Checklist of Mixed Methods Elements in a Submission for Advancing the Methodology of Mixed Methods Research. Journal of Mixed Methods Research, 13(4), 414-423. https://doi.org/10.1177/1558689819875832).

Other additions/changes are listed below:

Title: integrate the study design into the title, e.g.: “Older Adults Affected by Abuse: What about their health and social participation? A mixed methods study”

Line 38: a typo (healthcaer)

Line 136: a typo (one space is missing)

Introduction: Check the color of the text

Table 1: table format and layout to be reviewed

Table 1: Insert a column with the standard deviation

Table 2: table format and layout to be reviewed

Table 2: replace “Sub-category” with “Codes”, “Generic category” with “Sub-themes” or “Sub-categories” and “Main category” with “Themes” or “Categories”

Results: it is necessary to review the formatting of the text and subchapters (font size, italics, etc)

Results: the quotes inserted in the text must be easily identifiable (e.g. italic and centered) and the alphanumeric code of the participant who said each sentence must be reported.

Best regards.

Author Response

(The authors gave the same response as above.)
